# The effect of a digital targeted client communication intervention on pregnant women's worries and satisfaction with antenatal care in Palestine–A cluster randomized controlled trial

**Binyam Bogale[1,2], Kjersti Mørkrid[1], Eatimad Abbas[3], Itimad Abu Ward[3], Firas Anaya[4], Buthaina Ghanem[3], Taghreed Hijaz[5], Mervett Isbeih[3], Sally Issawi[3], Zaher A. S. Nazzal[4], Sharif E. Qaddomi[3], J. Frederik Frøen[1,2]\***

1 Division for Health Services, Global Health Cluster, Norwegian Institute of Public Health, Oslo, Norway, 2 Center for Intervention Science in Maternal and Child Health, University of Bergen, Bergen, Norway, 3 The Palestinian National Institute of Public Health, Irbid, Palestine, 4 Faculty of Medicine and Health Sciences, An-Najah National University, Nablus, Palestine, 5 The Palestinian Ministry of Health, Ramallah, Palestine

\* frederik.froen@fhi.no

**Data Availability Statement:** All relevant data are within the paper and its Supporting Information files.

## Abstract

### Background

The eRegCom cluster randomized controlled trial assesses the effectiveness of targeted client communication (TCC) via short message service (SMS) to pregnant women, from a digital maternal and child health registry (eRegistry) in Palestine, on improving attendance and quality of care. In this paper, we assess whether this TCC intervention could also have unintended consequences on pregnant women's worries, and their satisfaction with antenatal care (ANC).

### Methods

We interviewed a sub-sample of Arabic-speaking women attending ANC at public primary healthcare clinics, randomized to either the TCC intervention or no TCC (control) in the eRegCom trial, who were in 38 weeks of gestation and had a phone number registered in the eRegistry. Trained female data collectors interviewed women by phone from 67 intervention and 64 control clusters, after securing informed oral consent. The Arabic interview guide, pilot-tested prior to the data collection, included close-ended questions to capture the woman's socio-demographic status, agreement questions about their satisfaction with ANC services, and the 13-item Cambridge Worry Scale (CWS). We employed a non-inferiority study design and an intention-to-treat analysis approach.

### Results

A total of 454 women, 239 from the TCC intervention and 215 from the control arm participated in this sub-study. The mean and standard deviation of the CWS were 1.8 (1.9) for the

**Funding:** This study and the eRegistry research project are funded by the European Research Council [https://erc.europa.eu/] (grant agreement number: 617639, project title: A New Paradigm for Public Health Surveillance: Unlocking the Potential of Data to Empower Woman and Health Systems; project acronym, HEALTHMPOWR) and the Research Council of Norway [https://www.forskningsradet.no/en/] (grant agreement number: 234376, project title: Harmonized Reproductive Health Registry Communication Strategies: Using Health Data to Empower Women and Health Systems; and grant agreement number: 223269, project title: Center for Intervention Science in Maternal and Child health (CISMAC) [https://www.uib.no/en/cismac], University of Bergen, Norway). The funders had no role in study design, data collection and analysis, decision to publish, or preparation of the manuscript.

**Competing interests:** The authors have declared that no competing interests exist.

**Abbreviations:** ANC, Antenatal Care; CI, Confidence Intervals; CWS, Cambridge Worry Scale; ICC, Intra-cluster Correlation Coefficient; MCH, Maternal and Child Health; PHC, Primary Healthcare Center; PNIPH, Palestinian National Institute of Public Health; QID, Quality Improvement Dashboard; SD, Standard deviation; SMS, Short Message Service; TCC, Targeted Client Communication.

intervention and 2.0 (1.9) for the control arm. The difference in mean between the intervention and control arms was -0.16 (95% CI: -0.31 to -0.01) after adjusting for clustering, which was below the predefined non-inferiority margin of 0.3. Women in both groups were equally satisfied with the ANC services they received.

## Conclusion

The TCC intervention via SMS did not increase pregnancy-related worries among recipients. There was no difference in women's satisfaction with the ANC services between intervention and control arms.

## Introduction

Targeted client communication (TCC) using Short Message Service (SMS), is among the most common digital health interventions [1]. The most effective digital TCC interventions are co-designed with users, underpinned by behavior change theories, tested, and iteratively improved [1–3]. Pure appointment reminders have shown moderate effectiveness in improving attendance to maternity services [1,4,5]. However, digital health communication interventions tailored to the individual recipient are more likely to lead to behavior change compared with generic communication [6,7]. Health education and promotion messages via SMS can empower women to make informed health choices, which may contribute to a positive pregnancy experience [1,2,4,8,9]. While tailoring based on individual-level risk factors has advantages, it may also result in potential unintended consequences, such as, the triggering of worries among the message recipients. Documenting and preventing potential unintended consequences have generally been given little attention in the field of health education and promotion interventions [10].

TCC intervention studies often report the effectiveness as a main outcome, and seldom include a robust study design to assess its potential negative effects, such as adverse psychological outcomes and clients' satisfaction [1,11]. In their guidelines for digital interventions for health system strengthening, the World Health Organization (WHO) highlights the importance of assessing any unintended consequences of, and client's satisfaction with digital health interventions, in addition to the effectiveness among others [12,13].

Pregnancy is a period when women are more vulnerable to worries and anxiety, which are often highest in early and late stages of pregnancy [14,15]. The variabilities in the definition and the psychometric measurement tools used across studies, hamper the understanding of worries and anxiety in pregnancy [15–17]. Nevertheless, the psycho-social environment, and previous and current obstetric and medical status were among the risk-factors causing worries in pregnancy [14]. Generic antenatal health education and promotion utilizing digital health technology can reduce pregnancy-related concerns and worries [18–20], but there is limited information on the potential adverse effects [21].

In a client-centered maternity care model, women's satisfaction is an integral part of the quality of services [22,23]. Well-informed pregnant women are more likely to make informed health choices, and they are often satisfied with the antenatal care (ANC) services they receive. A well-designed TCC intervention using SMS, in addition to the routine antenatal education program, may improve women's satisfaction [20].

We have previously reported low effective coverage of essential interventions of ANC in Palestine, mainly attributed to untimely attendance in public primary healthcare center (PHC)

[24]. An electronic registry that includes systematic, uniform, and longitudinal client information entered at the point-of-care, such as the Maternal and Child Health (MCH) eRegistry in Palestine, provides a unique opportunity for tailored TCC via SMS to each woman to improve attendance [25–27]. We have developed a theory-based, co-designed, and user tested TCC intervention to pregnant women automated from the MCH eRegistry [25]. The aim of the TCC intervention was to increase the awareness of individual-level susceptibility to, and severity of, prioritized pregnancy-related high-risk conditions, specifically, anemia, hypertension, diabetes, and growth restriction, and thus improve timely attendances for screening and management of the high-risk conditions. This digital health intervention is under assessment for its effectiveness in a four-armed cluster randomized trial (eRegCom: Trial registration number: ISRCTN10520687).

Efforts to minimize potential worries were made during the design of the TCC intervention; however, we cannot exclude the potential increase in pregnancy-related worries. This might be particularly relevant for pregnant women receiving text messages with tailored information about their risk factors, such as high body mass index, high or low age, and a history of pregnancy complications; and the link to one of the prioritized pregnancy-related high risk conditions [25].

The objective of this sub-study of the eRegCom trial was to assess whether this TCC intervention via SMS, automated from the Palestinian MCH eRegistry, could affect pregnant women's worries and satisfaction with ANC services.

## Methods

### Trial design and participants

This was a non-inferiority two-armed parallel cluster randomized trial, sub-sampled from the four-armed eRegCom trial (Trial registration number: ISRCTN10520687), following the Consolidating Standards of Reporting Trials (CONSORT) criteria for cluster randomized trials [28] described in detail elsewhere [29]. In short, the four arms include one arm with Quality Improvement Dashboards (QID) for healthcare providers; one with TCC via SMS to women; one with both QID and TCC via SMS; and one control arm. The 138 clusters (one closed after randomization) in the eRegCom trial are public PHC offering both antenatal and postnatal care services, active users of the MCH eRegistry, and served 45 to 3000 new pregnancies in 2016.

For this sub-study, the TCC intervention arm (69 clusters) includes both arms with TCC intervention in the eRegCom trial, and the control arm (68 clusters) includes both arms without it. Additional inclusion criteria for this sub-study were that the women had registered a phone number in the eRegistry, were in the 38th week of gestation, and spoke Arabic.

### Intervention

The development process and content of the TCC intervention are described elsewhere [25]. In short, the TCC intervention in the eRegCom trial includes training of healthcare providers on how to enroll women in the TCC program, which sends a series of individualized and automated text messages. Routine clinical data captured by the healthcare provider at the point-of-care are applied in algorithms that identify the correct text message to each individual woman. The woman's name, the date of her next appointment and the name of the PHC are automatically inserted into one of the 56 unique predesigned text message templates stored in the library. The text messages that include information about one or two of the prioritized pregnancy-related high-risk conditions, are sent at the time these conditions are screened for, namely at the 16, 18–22, 24–28, 32- or 36-weeks' gestation routine ANC visits. Women receive

a welcome text message to the mobile number registered during the first ANC visit (booking) or any visit where they assent to take part in the text message program; one week, three days, and 24 hours before a scheduled appointment; 24 hours after a missed appointment; and 24 hours prior to an appropriate time window without any timely scheduled routine visit in the future.

## Data collection

We used the 13-item Cambridge Worry Scale (CWS) [30] which utilizes a six-point Likert-type scale ranging from 0 (not a worry) to 5 (extremely worried). In addition to the CWS, the interview guide included close-ended questions to capture the women's socio-demographic status, and agreement scale questions (0 = strongly disagree to 5 = strongly agree) about satisfaction with ANC services. Two researchers, fluent in English and Arabic, translated the English version to Arabic, which was back translated by a third individual prior to a pilot test in January 2019.

Four trained and experienced female data collectors conducted the phone interviews while being blinded to the allocation and primary outcome, fluent in the Arabic language and familiar with the local context, including ANC terminologies. The data collectors received a password-protected document with a list of eligible women and their registered phone numbers on a weekly basis. The lists were deleted after one week. A data manager oversaw the preparation and distribution of lists, including the safe storage of the allocation key. The data collectors entered the response in a pre-designed Google Form while interviewing, and the data quality and completeness were monitored daily. The data collectors tried to reach each woman on a maximum of three different occasions.

## Outcomes

The primary outcome of this sub-study was the difference in mean total CWS score between the intervention and control arms with the one-sided confidence interval (CI) considering the non-inferiority margin. The mean and standard deviation (SD) of the total 13-items CWS score were calculated for each study arm. A higher mean score suggests higher levels of worries. We categorized the 13-items into the four predefined components, namely socio-medical, socio-economic, health, and relationship [30–36].

We also measured pregnant women's satisfaction with ANC services, computed as the mean difference of each question on an agreement scale.

## Sample size

The power of the study was estimated using PASS software for sample size calculation for a non-inferiority cluster randomized trial design [37]. We hypothesized that there was no statistically significantly increased difference in the mean total CWS score between women in the intervention and control arms with a given non-inferiority margin. We were more than 90% powered to detect a non-inferiority margin of a one-point increase of every third questions on the CWS (corresponding to an increase of 0.3 in the CWS mean score), with standard deviation (SD) of 1.1, assumed intra-cluster correlation coefficient (ICC) of 0.01, a one-sided significance level of 2.5% for 137 clusters with equal cluster size of four. The SD and a pooled mean score of 1.5 [95% CI: 1.09–1.92] were computed from a meta-analysis of relevant literature and a pilot study (n = 41) conducted in the same study site in January 2019.

### Randomization

An independent statistician performed the randomization for the eRegCom trial [29], where PHCs were randomized to the TCC; QID; TCC and QID; or control arm with equal allocation. The randomization was stratified by the national implementation phase of the eRegistry, and constrained on laboratory availability, ultrasound availability and the size of the PHCs.

### Statistical methods

We applied an intention-to-treat analysis and used R software with the lme4 package for the Mixed Effect Linear Regression Model to consider the cluster effect for the computation of the difference in the mean total CWS score between the intervention and control arms. We used an unstructured covariance model to impose the fewest assumptions. In exploratory analyses, we found that this resulted in the best fitting model. We performed an exploratory analysis to assess any imbalances across the groups in potential confounders for the CWS like women's educational status, previous adverse pregnancy outcomes, age, and parity. We also used a previously established four factor structure for the CWS to analyze the unadjusted difference in mean scores. The ICC for the difference in mean CWS score and Cronbach's Alpha to see the internal consistency of the measurement were calculated. A statistician blinded for the allocation performed the final analysis.

### Ethics approval

The data collectors read out the information sheet and secured oral informed consent from all participants before conducting the phone interview. We obtained ethical clearance from the Helsinki Committee for Ethical Approval in Palestine (ref. no.: PHRC/HC/670/19) and an exemption from ethical review from the Regional Committee for Health Research Ethics (REK)—Section South East B, from Norway (ref.: REK sør-øst 139204) as health systems research falls outside of the mandate for ethical review in Norway.

## Results

We interviewed 239 women from 67 TCC intervention clusters, and 215 women from 64 control clusters in February and March 2020 (Fig 1). This was in total 83% of the calculated sample size. The data collection did not continue to the full sample size (estimated four women per cluster) as the TCC intervention was discontinued in March 2020 due to the COVID-19 situation in Palestine, and women in the intervention arm would no longer have received the full intervention after this point.

There were no statistically significant imbalances in background characteristics between the intervention and control arms. The majority of women were between 25–29 years of age (Table 1). About half of the women had a college or university level education, and only one woman reported no formal education. In control arm PHCs, 35% were primiparous and 74% of the women had their first ANC visit in the first trimester. About 80% attended regular ANC and 90% ultrasound services in private/Non-Governmental Organization/United Nations clinics, in addition to the public PHC they were registered to. More than 85% in each arm had their own mobile phone.

### Worries in pregnancy

Generally, women in the intervention clusters scored lower on the CWS than their counterparts in the control clusters, but the difference was not statistically significant. Items directly related to the pregnancy (giving birth, going to hospital, and internal examinations), along

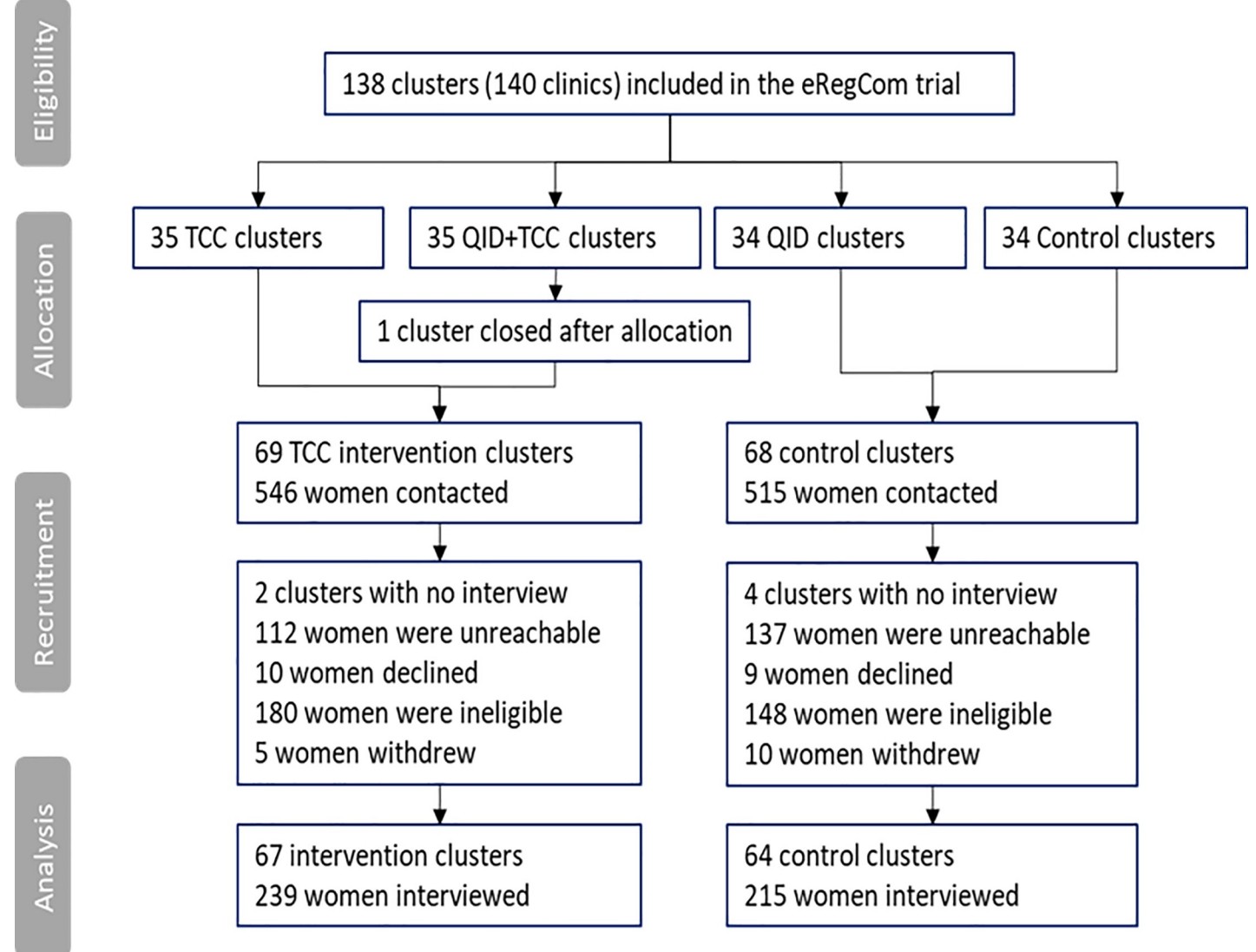

**Fig 1. Participant flow chart (CONSORT diagram).**

with items related to the baby had the highest scores of worries in both arms. Fig 2 presents a crude mean value per intervention and control arms without adjusting for clustering effect.

There was no item in the CWS that none of the women in our sample were not worried about, although the most frequent response was "no worry at all". The unadjusted total mean CWS score and SD were 1.8(1.9), and 2.0(1.9) in intervention and control arms respectively (Table 2). The previously established socio-medical and health factor structures comprised the highest worries in both arms. The reliability of the CWS was satisfactory (Cronbach's Alpha = 0.74).

After adjusting for the clustering effect, the difference in mean score was -0.16 (95%CI: -0.31 to -0.01), where the upper limit of the confidence interval was -0.01 (taking 2.5% confidence level on each end), which was lower than the predefined non-inferiority margin of 0.3 for the hypothesis testing. The ICC for the difference in mean CWS score was 0.01.

**Table 1. Background characteristics of the study participants in intervention and control arms.**

| Background characteristics | Intervention arm (clusters = 67, n = 239) | Control arm (clusters = 64, n = 215) |
|---|---|---|
| | n (%) | n (%) |
| **Women's age** | | |
| <20 | 21 (9) | 16 (8) |
| 20–24 | 57 (24) | 59 (28) |
| 25–29 | 84 (35) | 74 (35) |
| 30–34 | 49 (21) | 56 (26) |
| 35–39 | 22 (9) | 6 (3) |
| ≥ 40 | 5 (2) | 3 (1) |
| **Educational status** | | |
| Primary | 3 (1) | 1 (1) |
| Secondary | 114 (48) | 113 (53) |
| College or University | 117 (49) | 96 (45) |
| After college or University | 5 (2) | 4 (2) |
| No formal education | 0 (0) | 1 (1) |
| **Work status** | | |
| Work outside the home | 26 (11) | 33 (16) |
| No work outside the home | 211 (89) | 180 (85) |
| **Parity** | | |
| Primipara | 69 (29) | 74 (35) |
| Multipara | 169 (71) | 140 (65) |
| **Gestational age at booking** | | |
| < 4 months | 160 (70) | 160 (74) |
| 4–6 months | 67 (28) | 46 (21) |
| > 6 months | 12 (5) | 8 (4) |
| **Referral to high-risk clinics** | | |
| Yes | 35 (15) | 31 (14) |
| No | 204 (85) | 184 (86) |
| **Attended ANC in another clinic** | | |
| Yes | 203 (85) | 170 (79) |
| No | 36 (15) | 45 (21) |
| **Attended for ultrasound in another clinic** | | |
| Yes | 211 (90) | 189 (88) |
| No | 23 (10) | 26 (12) |

n: Number of women.

## Satisfaction with ANC services

In both groups, three of four women moderately or strongly agreed to the statement "I am satisfied with the antenatal care services I have received", and an equal majority responded that they would come back to the same PHC, if become pregnant again, and would recommend it to others. Most of the pregnant women were moderately or strongly satisfied with the information and communication from their care providers, and there was no difference between the groups (Table 3).

## Discussion

We have demonstrated that pregnant women receiving digital tailored text messages with health content during pregnancy, were not more worried compared to women in the control

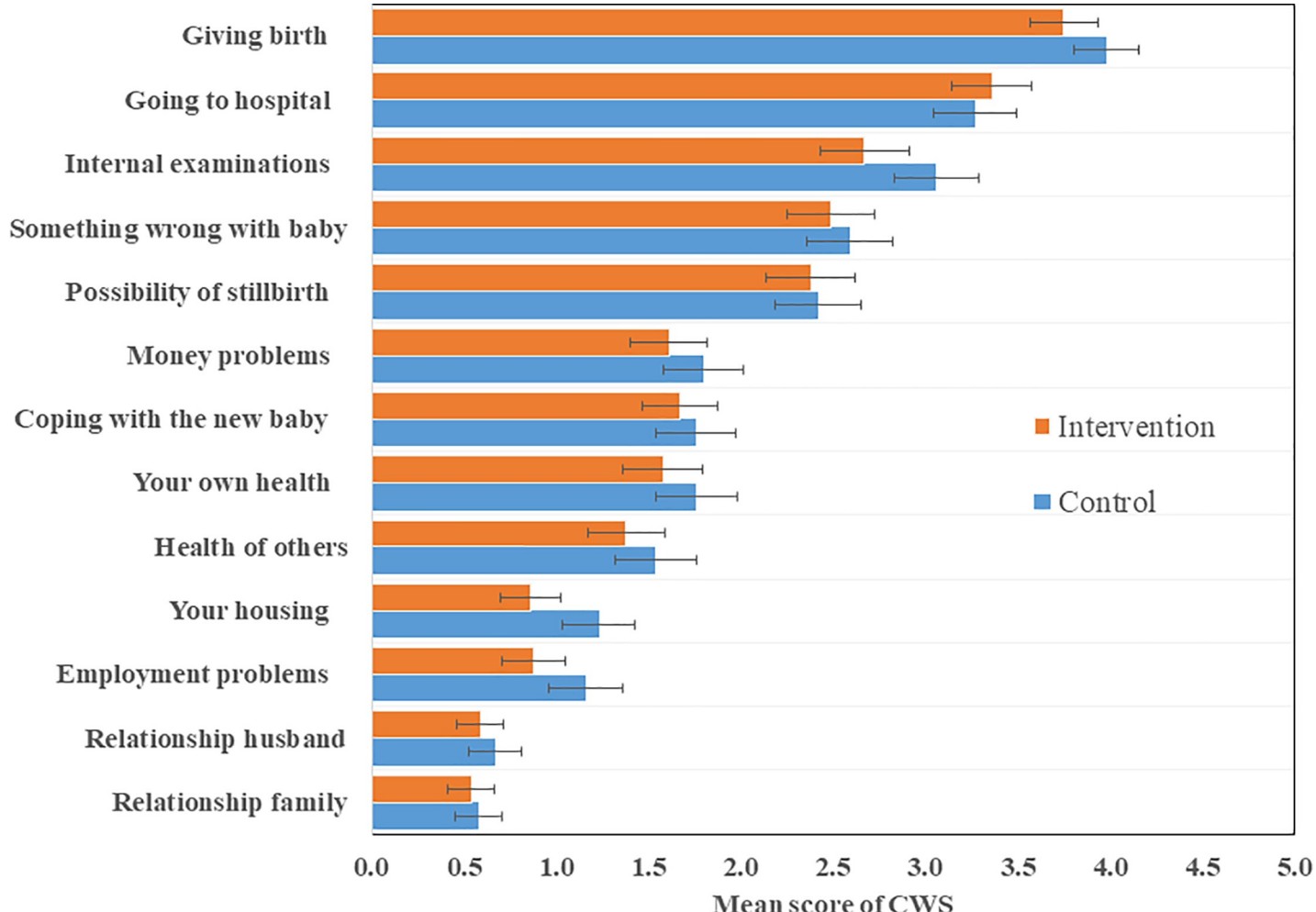

**Fig 2. Ranking of women's mean worry score with 95% confidence interval for intervention and control arms.**

clusters, measured by the CWS. Even though not statistically significant, both the total and single item mean CWS scores were consistently lower for women in the TCC intervention arm compared to their counterparts in the control arm. Women in both arms were equally and generally satisfied with the ANC services they received from their public PHCs.

Most effectiveness trials of digital health interventions fail to report potential unintended consequences of the interventions [1,4,5]. This research gap is highlighted by WHO in its first digital health guideline [13]. The majority of health messaging interventions using mobile technologies, have up until now, included generic health promotion content [4]. Such content is generally acceptable and have most likely been considered as safe, which might explain the low attention towards unintended consequences of public health intervention in general [38], and health promotion in particular [10]. Precautions in composing individually tailored messages, which are increasing, are warranted to avoid harmful consequences. Furthermore, the communication of individualized risk factors to pregnant women needs special attention [39]. Co-designing theory-based interventions in an iterative process with users is recommended to secure the safety and effectiveness of messages [1–5]. We have followed these recommendations, and findings from a systematic review recommending identifying the sender, using a

**Table 2. Unadjusted summaries of the Cambridge Worry Scale for the total sample, and across study arms.**

| Cambridge Worry Scale items | Freq (n) | Level of worry (%)[e] | | | | | | Intervention (clusters = 67 n = 239) | Control (clusters = 64 n = 215) | Difference in Mean score (95% CI) |
|---|---|---|---|---|---|---|---|---|---|---|
| | | 0 | 1 | 2 | 3 | 4 | 5 | Mean (SD) | Mean (SD) | |
| *Your housing* | 449 | 53 | 26 | 1 | 10 | 6 | 5 | 0.9 (1.3) | 1.2 (1.6) | -0.4 (-0.7, -0.1) |
| *Money problems* | 451 | 38 | 21 | 2 | 19 | 12 | 8 | 1.6 (1.7) | 1.8 (1.8) | -0.2 (-0.5, 0.1) |
| *Your relationship with your husband/ partner* | 448 | 62 | 27 | 1 | 6 | 2 | 1 | 0.6 (1.0) | 0.7 (1.1) | -0.1 (-0.3, 0.1) |
| *Your relationship with your family and friends* | 449 | 64 | 28 | 0 | 4 | 2 | 2 | 0.5 (1.1) | 0.6 (1.0) | -0.0 (-0.2, 0.2) |
| *Your own health* | 446 | 41 | 19 | 2 | 15 | 15 | 8 | 1.6 (1.8) | 1.8 (1.8) | -0.2 (-0.5, 0.2) |
| *The health of someone close to you* | 449 | 45 | 25 | 1 | 8 | 14 | 7 | 1.4 (1.7) | 1.5 (1.8) | -0.2 (-0.5, 0.2) |
| *Employment problems* | 451 | 55 | 25 | 1 | 7 | 7 | 5 | 0.9 (1.4) | 1.2 (1.6) | -0.3 (-0.6, 0.0) |
| *The possibility of something being wrong with the baby* | 447 | 26 | 14 | 2 | 14 | 27 | 17 | 2.5 (2.0) | 2.6 (1.9) | -0.1 (-0.5, 0.3) |
| *The possibility of stillbirth* | 447 | 28 | 16 | 2 | 13 | 21 | 19 | 2.4 (2.0) | 2.4 (1.9) | 0.0 (-0.4, 0.3) |
| *Going to hospital* | 448 | 16 | 9 | 1 | 10 | 30 | 34 | 3.4 (1.8) | 3.3 (1.8) | 0.1 (-0.3, 0.4) |
| *Internal examinations* | 451 | 21 | 13 | 2 | 13 | 24 | 26 | 2.7 (2.0) | 3.1 (1.8) | -0.4 (-0.7, 0.0) |
| *Giving birth* | 449 | 7 | 4 | 3 | 12 | 29 | 45 | 3.8 (1.5) | 4.0 (1.4) | -0.2 (-0.5, 0.1) |
| *Coping with the new baby* | 451 | 36 | 22 | 6 | 16 | 13 | 8 | 1.7 (1.7) | 1.8 (1.7) | -0.1 (-0.4, 0.2) |
| Total CWS (13-items) | | | | | | | | **1.8 (1.9)** | **2.0 (1.9)** | **-0.2 (-0.3, -0.1)** |
| Socio-medical [a] | | | | | | | | 2.9 (1.9) | 3.0 (1.9) | -0.2 (-0.3, 0.0) |
| Socio-economic [b] | | | | | | | | 1.1 (1.5) | 1.4 (1.7) | -0.3 (-0.5, -0.1) |
| Health [c] | | | | | | | | 2.0 (1.9) | 2.1 (1.9) | -0.1 (-0.3, 0.1) |
| Relationship [d] | | | | | | | | 0.6 (1.0) | 0.6 (1.1) | -0.1 (-0.2, 0.1) |

[a](Worry of giving birth, internal examinations, going to hospital, coping with the new baby),

[b](money problems, housing problems, employment problems),

[c](possibility of stillbirth, something wrong with the baby, own health, others' health,

[d](relationship with the family, relationship with partner); Level of worry (0 = not a worry, 5 = extremely worried) n: Number of women.

positively framed tone, and including content with solutions in a structured and focused manner [40].

Maternal age, previous adverse pregnancy outcomes, education, and employment status are known predictors of worries in pregnancy [14]. These can also affect the acceptance and understanding of the TCC intervention, hence be potential confounders to the main outcome of the study. However, we did not identify any statistically significant imbalances of these variables between intervention and control arms, and they were therefore not included in the final model. We took the cluster effect into account, but it did not markedly change the point estimate or the confidence interval of the estimate.

The mean CWS score for pregnant women in Palestine did not differ from other studies reported using the CWS [30–35]. The items, such as, giving birth, hospital visits, internal examinations, something might be wrong with the baby, and the possibility of stillbirth, were in line with other studies, recoded with the highest mean scores [32]. Similarly, pregnant women in both the intervention and control arms were more worried about the socio-medical and health components compared to the socio-economic and relationship components of the CWS. We did not aim to validate the CWS (has not been validated in Arabic language); however, the good internal consistency score (Cronbach's alpha = 0.74) might indicate that the CWS can be used for pregnant women in Palestine. Since this study is a non-inferiority trial, it

**Table 3. Women's satisfaction with ANC services among TCC intervention and control arms in Palestine.**

| | Trial arm | Total | Level of agreement (%)[a] | | | | | | Mean (SD) | Mean Difference (95% CI) [b] |
|---|---|---|---|---|---|---|---|---|---|---|
| | | | 0 | 1 | 2 | 3 | 4 | 5 | | |
| I am satisfied with the antenatal care service I have received | Control | 214 | 3 | 2 | 3 | 18 | 33 | 41 | 4.0 (1.2) | 0.1(-0.1, 0.3) |
| | Intervention | 237 | 3 | 0 | 3 | 16 | 33 | 46 | 4.1 (1.1) | |
| I would recommend the services to a friend | Control | 214 | 3 | 5 | 1 | 6 | 42 | 44 | 4.1 (1.2) | 0.1 (-0.1, 0.3) |
| | Intervention | 235 | 2 | 3 | 0 | 7 | 43 | 45 | 4.2 (1.0) | |
| I would come back if I become pregnant again | Control | 212 | 3 | 3 | 1 | 5 | 35 | 53 | 4.2 (1.2) | 0.1 (-0.1, 0.3) |
| | Intervention | 237 | 2 | 3 | 1 | 5 | 38 | 52 | 4.3 (1.0) | |
| I am always confident of when my next antenatal care visit is | Control | 214 | 3 | 1 | 1 | 6 | 27 | 64 | 4.4 (1.0) | 0.0 (-0.2, 0.1) |
| | Intervention | 237 | 2 | 3 | 0 | 3 | 31 | 60 | 4.4 (1.0) | |
| The health staff take my questions and concerns seriously | Control | 210 | 2 | 2 | 1 | 9 | 37 | 49 | 4.2 (1.0) | 0.1 (-0.1, 0.3) |
| | Intervention | 236 | 2 | 8 | 0 | 6 | 34 | 56 | 4.4 (1.0) | |
| I am well informed about the purpose of the tests the health staff run | Control | 213 | 3 | 8 | 5 | 6 | 38 | 40 | 3.9 (1.4) | 0.1 (-0.1, 0.4) |
| | Intervention | 235 | 3 | 8 | 3 | 4 | 36 | 46 | 4.0 (1.3) | |
| I am well informed of when (gestational age) to do the tests | Control | 213 | 5 | 6 | 4 | 5 | 39 | 42 | 3.9 (1.4) | 0.1 (-0.2, 0.3) |
| | Intervention | 235 | 3 | 8 | 3 | 5 | 40 | 43 | 4.0 (1.3) | |

SD: Standard Deviation; CI: Confidence Interval;

[a] Level of agreement (0 = strongly disagree, 1 = moderately disagree, 2 = slightly disagree, 3 = slightly agree, 4 = moderately agree, 5 = strongly agree);

[b] unadjusted for the cluster effect.

does not intend to measure any benefit of the TCC intervention in terms of reducing pregnancy-related worries.

Due to the closure of PHCs and disruption of the TCC intervention in response to the COVID-19 pandemic, we ended the data collection period earlier than planned to ensure that we only included women who had been eligible for the full intervention. This resulted in the loss of six clusters and we only reached 83% of the planned sample size. However, we have no reason to believe that this has affected the main findings, nor that the emerging epidemic of COVID-19 in Palestine unequally affected the worries among women across arms. Our ethical clearance did not cover merging the sub-sample data with the data in the eRegistry. We have therefore no information about the number of timely scheduled ANC visits, the number of text messages, nor which text messages each woman in the sub-sample have received. Two in three women attended their first ANC visit in their first trimester and would be eligible to receive the full sequence of messages, a service that started in June 2019.

We interviewed women towards the end of their pregnancy and prior to delivery to ensure that they had been eligible for the full intervention and avoid any recall bias. Women who delivered or had an abortion prior to 38 weeks of gestation were not included. Our trained data collectors were not affiliated with the primary healthcare services to reduce the social desirability bias. The data collectors were blinded for the allocation to reduce the potential selection bias. The main strength of this sub-study was the study methodology including the design, the use of the validated CWS, and that we conducted a pilot test of the entire study tool.

## Conclusion

Our TCC intervention via SMS to improve attendance to ANC did not have any unintended consequences in increasing pregnancy-related worries among recipients. There was no difference in women's satisfaction with the ANC services between intervention and control arms.

## Supporting information

**S1 Checklist.**
(DOCX)

**S1 Dataset.**
(XLT)

## Author Contributions

**Conceptualization:** Binyam Bogale, Kjersti Mørkrid, Buthaina Ghanem, Mervett Isbeih, Zaher A. S. Nazzal, J. Frederik Frøen.

**Data curation:** Binyam Bogale, Kjersti Mørkrid, Eatimad Abbas, Itimad Abu Ward, Firas Anaya, Taghreed Hijaz, Mervett Isbeih, Sally Issawi, Sharif E. Qaddomi.

**Formal analysis:** Binyam Bogale, Sharif E. Qaddomi.

**Funding acquisition:** J. Frederik Frøen.

**Investigation:** Binyam Bogale, Kjersti Mørkrid, Eatimad Abbas, Itimad Abu Ward, Firas Anaya, Buthaina Ghanem, Taghreed Hijaz, Mervett Isbeih, Sally Issawi, J. Frederik Frøen.

**Methodology:** Binyam Bogale, Kjersti Mørkrid, Firas Anaya, Buthaina Ghanem, Taghreed Hijaz, Mervett Isbeih, Zaher A. S. Nazzal, J. Frederik Frøen.

**Project administration:** Binyam Bogale, Kjersti Mørkrid, Eatimad Abbas, Itimad Abu Ward, J. Frederik Frøen.

**Supervision:** Kjersti Mørkrid, Itimad Abu Ward, Buthaina Ghanem, Mervett Isbeih.

**Writing – original draft:** Binyam Bogale.

**Writing – review & editing:** Binyam Bogale, Kjersti Mørkrid, Eatimad Abbas, Itimad Abu Ward, Firas Anaya, Buthaina Ghanem, Taghreed Hijaz, Mervett Isbeih, Sally Issawi, Zaher A. S. Nazzal, Sharif E. Qaddomi, J. Frederik Frøen.

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
