## [Decision Letter · Decision Letter 0]

29 Sep 2020

PONE-D-20-20469

The effect of a digital targeted client communication intervention on pregnant women’s worries and satisfaction with antenatal care in Palestine – a cluster randomized controlled trial

PLOS ONE

Dear Dr. Frøen,

Thank you for submitting your manuscript to PLOS ONE. After careful consideration, we feel that it has merit but does not fully meet PLOS ONE’s publication criteria as it currently stands. Therefore, we invite you to submit a revised version of the manuscript that addresses the points raised during the review process.

All the three reviewers have raised pertinent issues that I agree with and must be addressed before the manuscript can be reconsidered for publication.

We look forward to receiving your revised manuscript.

Kind regards,

Godfrey Biemba, MBChB, M.Sc

Academic Editor

PLOS ONE

Journal Requirements:

Reviewers' comments:

Reviewer's Responses to Questions

**Comments to the Author**

1. Is the manuscript technically sound, and do the data support the conclusions?

Reviewer #1: Yes

Reviewer #2: Yes

Reviewer #3: No

2. Has the statistical analysis been performed appropriately and rigorously? 

Reviewer #1: Yes

Reviewer #2: Yes

Reviewer #3: No

3. Have the authors made all data underlying the findings in their manuscript fully available?

Reviewer #1: Yes

Reviewer #2: No

Reviewer #3: Yes

4. Is the manuscript presented in an intelligible fashion and written in standard English?

Reviewer #1: Yes

Reviewer #2: Yes

Reviewer #3: Yes

5. Review Comments to the Author

Reviewer #1: It is a nicely written manuscript. Just a few comments. The outcome of the study is presented as "improved attendance and quality of care" yet the AIM of the substudy is stated - to assess unintended consequences of the TCC intervention on pregnant women's worries, and their satisfaction with antenatal care. In Ln 75 the statement is made that the authors have developed a theory based, co-designed and used and tested TCC, but it sounds as if this was a secondary analysis of existing data. If so, this needs to be clarified....or consider rephrasing along the lines of "in this secondary analysis, data using the theory-based ...."

The abstract states that the primary reason for the TCC was to improve the quality of care yet the methods section goes into a lot of information about increasing awareness, improving attendance...were all these components assessed? Please align the abstract with the paper.

Be careful to use 'past-tense' terminology throughout the paper, Ln 106, 'were' applied, not 'are' applied, Ln 108, 'were' applied not 'are' applied.

Ln 138, Difference of 'each question' not 'each questions.'

Ln 245 - Validate instead of 'validating'

Nice statement about taking precautions in composing individualized messages and giving special attention to messaging for pregnant women.

Reviewer #2: A non-inferiority designed substudy was conducted to assess targeted client communication (TCC) intervention to no TCC (control) on pregnant women’s worries and satisfaction with antenatal care.

Minor revisions:

1- Line 156: Typographical error: Model.

2- Line 156: State the underlying covariance structure used in the mixed effect model and the criteria for selecting it.

3- Table 1: Include p-values to compare background characteristics between the intervention and control arms. Additionally, in the statistical method section, indicate the statistical methods used to estimate the p-values.

4- Tables 2 and 3: Provide an interpretation for each level of worry and level of agreement. Would these levels be considered a likert scale? Please clarify.

Reviewer #3: Dear Authors

1. The problem for intervention is not clear. I cloud not understand what is the problem of Palestinen women in antenatal care?

2. There is no relation between intervention with outcome assessment tool. The Authors conducted a digital targeted client communication intervention with SMS to attended for reeving prenatal care. After intervention they assessed the worries?

What was the content of message to reduction pf worries?

3. The authors concluded that "The TCC intervention via SMS did not increase pregnancy related worries among recipients"

I think that the authors did not select a suitable assessment tool for the intervention.

4. The inclusion and exclusion criteria is not written well.

5. The blindness and masking?

6. Fig participants is not clear. At initiation of study four arms and then two arm? Why?

I overall, I was confused. I cloud not find a suitable deification of problems, the intervention,, and assessment tool. I do not understand that why the main results of the trial is factors of worries of pregnant women?

6. PLOS authors have the option to publish the peer review history of their article (what does this mean?). If published, this will include your full peer review and any attached files.

Reviewer #1: No

Reviewer #2: No

Reviewer #3: No

---

## [Author Response · Author response to Decision Letter 0]

12 Nov 2020

Reviewer #1:  

It is a nicely written manuscript. Just a few comments. The outcome of the study is presented as "improved attendance and quality of care" yet the AIM of the sub-study is stated - to assess unintended consequences of the TCC intervention on pregnant women's worries, and their satisfaction with antenatal care. 

Our response: 

Thank you for identifying this. The statement was referring to the primary outcomes of the effectiveness trial on which this sub-study is based. This paper presents an additional data collection and analysis with the aim of assessing any unintended consequences of the intervention. We assessed whether women’s worries during pregnancy, and their satisfaction with the antenatal care service, could be negatively affected by the intervention. 

We have clarified this in statements in line 25 and 84. 

In Ln 75 the statement is made that the authors have developed a theory based, co-designed and used and tested TCC, but it sounds as if this was a secondary analysis of existing data. If so, this needs to be clarified....or consider rephrasing along the lines of "in this secondary analysis, data using the theory-based ...." 

Our response: 

This relates to the same unclarity as above. This paper represents a new data collection to assess any unintended consequences of a trial, which had the primary objective of improving attendance and quality of care. To clarify our point, we added sentence with reference to previous study in line 74-75, and rephrased the sentence in line 84. 

The abstract states that the primary reason for the TCC was to improve the quality of care yet the methods section goes into a lot of information about increasing awareness, improving attendance...were all these components assessed? Please align the abstract with the paper. 

Our response: 

Thank you for pointing out this for clarification. We have revised the abstract accordingly. Lines 25 and 137 

Be careful to use 'past-tense' terminology throughout the paper, Ln 106, 'were' applied, not 'are' applied, Ln 108, 'were' applied not 'are' applied. 

Ln 138, Difference of 'each question' not 'each questions.' 

Ln 245 - Validate instead of 'validating' 

Our response: thank you or identifying the errors. We corrected them. 

Nice statement about taking precautions in composing individualized messages and giving special attention to messaging for pregnant women. 

Thank you. This was our rationale in conducting this sub-study. 

Reviewer #2: A non-inferiority designed substudy was conducted to assess targeted client communication (TCC) intervention to no TCC (control) on pregnant women’s worries and satisfaction with antenatal care. 

Minor revisions: 

1- Line 156: Typographical error: Model. 

Our response: 

Thank you for identifying the error. Corrected. 

2- Line 156: State the underlying covariance structure used in the mixed effect model and the criteria for selecting it. 

Our response: 

Thank you for your suggestion. Now we have added more text into the methods section, line 162-164 

3- Table 1: Include p-values to compare background characteristics between the intervention and control arms. Additionally, in the statistical method section, indicate the statistical methods used to estimate the p-values. 

Our response: 

Thank you for raising this point. We believe we will leave this for the Editors to decide as a matter of journal policy. We have followed the CONSORT statement that recommends not to report significance tests of baseline characteristics (“Standard errors and confidence intervals [and therefore P-values] are not appropriate)”. If the Editors prefer the inclusion of significance tests, we will be happy to provide them. 

4- Tables 2 and 3: Provide an interpretation for each level of worry and level of agreement. Would these levels be considered a likert scale? Please clarify. 

Our response: 

Yes, both for worries and satisfaction, these are Likert-type scales where the respondents rate their agreement/disagreement with statements. We have added interpretation of the levels in the footnotes under Table 2 and 3. 

Reviewer #3: Dear Authors 

1. The problem for intervention is not clear. I could not understand what is the problem of Palestinian women in antenatal care? 

Our response: 

Thank you for the question. We have now added a sentence to indicate the gap our intervention intended to fill in lines 74-75, referring to a previous assessment of attendance and quality of care in Palestine. 

2. There is no relation between intervention with outcome assessment tool. The Authors conducted a digital targeted client communication intervention with SMS to attended for receiving prenatal care. After intervention they assessed the worries? 

Our response: 

Please see responses to reviewer #1. 

What was the content of message to reduction of worries? 

Our response: 

Our TCC intervention was aimed at improving attendance, not specifically reducing worries. Increased worries would be an unintended consequence of messaging to increase attendance. As we have reported in detail elsewhere, this intervention included messages that focused on potential risks in pregnancy, and how attendance to ANC could reduce risks. We wanted to study whether this kind of messages could affect worries in pregnancy. 

3. The authors concluded that "The TCC intervention via SMS did not increase pregnancy related worries among recipients" 

I think that the authors did not select a suitable assessment tool for the intervention. 

Our response: 

Please refer to comments above. The assessment tool, the Cambridge Worry Scale has been validated in multiple settings. 

4. The inclusion and exclusion criteria is not written well. 

Our response: 

For the overall trial criteria, we have included the reference to the protocol paper where detailed eligibility criteria were presented. For the specific inclusion and exclusion criteria for the additional data collection of this sub-study, we have clarified in lines 104-106 

5. The blindness and masking? 

Our response: 

Thank you for raising this question. For the data collection and analysis, we have stated that the data collectors were blinded for allocation and primary outcome (line 128), and the statistician was blinded for allocation (line 169). Otherwise, the nature of the intervention does not allow for blinding and masking participants. 

6. Fig participants is not clear. At initiation of study four arms and then two arm? Why? 

I overall, I was confused. I cloud not find a suitable deification of problems, the intervention,, and assessment tool. I do not understand that why the main results of the trial is factors of worries of pregnant women 

Our response: 

Please refer back to previous responses. How the participant flow chart came about is additionally explained in lines 95-106.

---

## [Decision Letter · Decision Letter 1]

17 Feb 2021

PONE-D-20-20469R1

The effect of a digital targeted client communication intervention on pregnant women’s worries and satisfaction with antenatal care in Palestine – a cluster randomized controlled trial

PLOS ONE

Dear Dr. Frøen,

Thank you for submitting your manuscript to PLOS ONE. After careful consideration, we feel that it has merit but does not fully meet PLOS ONE’s publication criteria as it currently stands. Therefore, we invite you to submit a revised version of the manuscript that addresses the points raised during the review process.

We look forward to receiving your revised manuscript.

Kind regards,

Dani Zoorob, MD MHA MBA

Academic Editor

PLOS ONE

Additional Editor Comments (if provided):

Thank you for the edits. There are just a few more minor edits please.

Line 28 - add a comma after the word clinics, and change the word whom to who

Line 31 - please add a dash between pilot and tested

Line 32-33 - please replace the semicolons with commas

Line 42 - please add a hyphen between the words pregnancy and related (and other places in the whole document, such as line 66)

Line 61 - I would suggest adding the wore 'more' just before the word 'vulnerable'

Line 70 - Please add a hyphen between well and designed

Line 76 - Please add a hyphen between theory and based

Line 88-90 - Please change the color of the wording to match the text

Line 96 - Please place a comma after PHC

Line 117 - Please switch the word 'in' to 'which utilizes'

Line 119 - Please use the word womEn instead of womAn

Question - has the CWS in Arabic been validated?

Line 123 - Please switch the word 'for' to 'to'

Line 123-125 Please move the phrase 'conducted the phone interviews' to be placed after the word collectors, and then add the phrase 'while being' before the term 'blinded'

Line 125 - Please ad a hyphen between password and protected

Line 135 - Please replace the word reflects with the word suggests

Line 158 - Please switch the word structure to model

Line 161 - Please remove the S from the word Factors

Line 190 - please correct the statement (English grammar) as there are multiple double negatives.

Line 245 - please place a semicolon before the word 'however'

Line 248 - please place a dash between the words pregnancy and related

Line 263 - please remove the word 'strong'

Line 264 - please delete the word 'statistics'

Reviewers' comments:

Reviewer's Responses to Questions

**Comments to the Author**

1. If the authors have adequately addressed your comments raised in a previous round of review and you feel that this manuscript is now acceptable for publication, you may indicate that here to bypass the “Comments to the Author” section, enter your conflict of interest statement in the “Confidential to Editor” section, and submit your "Accept" recommendation.

Reviewer #2: All comments have been addressed

Reviewer #3: All comments have been addressed

7. PLOS authors have the option to publish the peer review history of their article (what does this mean?). If published, this will include your full peer review and any attached files.

Reviewer #2: No

Reviewer #3: No

2. Is the manuscript technically sound, and do the data support the conclusions?

Reviewer #3: Yes

3. Has the statistical analysis been performed appropriately and rigorously? 

Reviewer #3: Yes

4. Have the authors made all data underlying the findings in their manuscript fully available?

Reviewer #3: Yes

5. Is the manuscript presented in an intelligible fashion and written in standard English?

Reviewer #3: Yes

6. Review Comments to the Author

Reviewer #3: Dear Authors

Thank you for following the suggestions well. I think that the paper is improved very well.

---

## [Author Response · Author response to Decision Letter 1]

19 Feb 2021

Response to the academic editor

Thank you for the comments and edits. We accepted all your comments and recommendations for edits. 

Regarding your question about validation of CWS in Arabic language, we added a phrase “(has not been validated in Arabic language)” in Line #250. 

Best regards,

---

## [Editor Report · Decision Letter 2]

8 Mar 2021

PONE-D-20-20469R2

The effect of a digital targeted client communication intervention on pregnant women’s worries and satisfaction with antenatal care in Palestine – a cluster randomized controlled trial

PLOS ONE

Dear Dr. Frøen,

Thank you for submitting your manuscript to PLOS ONE. After careful consideration, we feel that it has merit but does not fully meet PLOS ONE’s publication criteria as it currently stands. Therefore, we invite you to submit a revised version of the manuscript that addresses the points raised during the review process.

We look forward to receiving your revised manuscript.

Kind regards,

Dani Zoorob, MD MHA MBA

Academic Editor

PLOS ONE

Journal Requirements:

Additional Editor Comments (if provided):

Good morning and thank you for the edits.

The one component still missing in the document that was overlooked in the previous communication was modification of the figure titles.

Please modify the Figure 1 title to contain the verbiage of CONSORT diagram, for example Figure 1. Participant flow chart (CONSORT diagram).

Also, please add to the Trial Design and Participants the concept that the study setup followed the CONSORT criteria. for example, Line 94 'This is was a non-inferiority two-armed parallel cluster randomized trial, sub-sampled from the four-armed eRegCom trial (Trial registration number: ISRCTN10520687), set up based CONSORT criteria, with full study details described elsewhere'

---

## [Author Response · Author response to Decision Letter 2]

8 Mar 2021

Response to the academic editor

Thank you for the comments and edits. We accepted all your comments and recommendations for edits as indicated in with track changes in the manuscript. 

Best regards,

---

## [Editor Report · Decision Letter 3]

24 Mar 2021

The effect of a digital targeted client communication intervention on pregnant women’s worries and satisfaction with antenatal care in Palestine – a cluster randomized controlled trial

PONE-D-20-20469R3

Dear Dr. Frøen,

We’re pleased to inform you that your manuscript has been judged scientifically suitable for publication and will be formally accepted for publication once it meets all outstanding technical requirements.

Kind regards,

Dani Zoorob, MD MHA MBA

Academic Editor

PLOS ONE
---

## [Editor Report · Acceptance letter]

13 Apr 2021

PONE-D-20-20469R3 

The effect of a digital targeted client communication intervention on pregnant women’s worries and satisfaction with antenatal care in Palestine – a cluster randomized controlled trial 

Dear Dr. Frøen:

I'm pleased to inform you that your manuscript has been deemed suitable for publication in PLOS ONE. Congratulations! Your manuscript is now with our production department. 

Kind regards, 

on behalf of

Dr Dani Zoorob 

Academic Editor

PLOS ONE